# School Problems and School Support for Children with Narcolepsy: Parent, Teacher, and Child Reports

**DOI:** 10.3390/ijerph20065175

**Published:** 2023-03-15

**Authors:** Karin Janssens, Pauline Amesz, Yvonne Nuvelstijn, Claire Donjacour, Danielle Hendriks, Els Peeters, Laury Quaedackers, Nele Vandenbussche, Sigrid Pillen, Gert Jan Lammers

**Affiliations:** 1Sleep-Wake Centre, Stichting Epilepsie Instellingen Nederland (SEIN), 8025 BV Zwolle, The Netherlands; 2Sleep-Wake Centre, Stichting Epilepsie Instellingen Nederland (SEIN), 2103 SW Heemstede, The Netherlands; 3LWOE, 2142 ED Cruquius, The Netherlands; 4Sleeping Center, Medical Centre Haaglanden, 2512 VA The Hague, The Netherlands; 5Department of Child Neurology, Juliana Children’s Hospital-Haga Teaching Hospital, 2545 AA The Hague, The Netherlands; 6Center for Sleep Medicine, Kempenhaeghe, 5591 VE Heeze, The Netherlands; 7Department of Industrial Design, Eindhoven University of Technology, 2612 AZ Eindhoven, The Netherlands; 8Noorderhart, Mariaziekenhuis, 3900 Pelt, Belgium; 9Kinderslaapexpert BV (Pediatric Sleep Expert Ltd.), 6585 KK Mook, The Netherlands; 10Department of Neurology, Leiden University Medical Center, 2333 ZA Leiden, The Netherlands

**Keywords:** narcolepsy type 1, school functioning, quality of life, school support, children

## Abstract

Objective: To assess problems faced by children with type 1 narcolepsy (NT1) at school and obtain insight into potential interventions for these problems. Methods: We recruited children and adolescents with NT1 from three Dutch sleep-wake centers. Children, parents, and teachers completed questionnaires about school functioning, interventions in the classroom, global functioning (DISABKIDS), and depressive symptoms (CDI). Results: Eighteen children (7–12 years) and thirty-seven adolescents (13–19 years) with NT1 were recruited. Teachers’ most frequently reported school problems were concentration problems and fatigue (reported by about 60% in both children and adolescents). The most common arrangements at school were, for children, discussing school excursions (68%) and taking a nap at school (50%) and, for adolescents, a place to nap at school (75%) and discussing school excursions (71%). Regular naps at home on the weekend (children 71% and adolescents 73%) were more common than regular naps at school (children 24% and adolescents 59%). Only a minority of individuals used other interventions. School support by specialized school workers was associated with significantly more classroom interventions (3.5 versus 1.0 in children and 5.2 versus 4.1 in adolescents) and napping at school, but not with better global functioning, lower depressive symptom levels, or napping during the weekends. Conclusions: Children with NT1 have various problems at school, even after medical treatment. Interventions to help children with NT1 within the classroom do not seem to be fully implemented. School support was associated with the higher implementation of these interventions. Longitudinal studies are warranted to examine how interventions can be better implemented within the school.

## 1. Introduction

Type 1 narcolepsy (NT1) is a chronic neurological sleep disorder characterized by excessive daytime sleepiness (EDS), which frequently induces problems with sustained attention. Emotion-triggered cataplexy, sleep paralysis, hypnagogic hallucinations, and disrupted nocturnal sleep are also common [1]. EDS is typically the first symptom to appear, followed by cataplexy up to several years later [1]. Onset is generally during adolescence but can also occur in childhood. Prevalence is 0.02–0.05% in western countries and an increase in annual incidence has been found among children [2]. NT1 is distinguished from Type 2 narcolepsy (NT2); in NT2, EDS is present, but cataplexy or low hypocretin levels are, in contrast to NT1, absent.

Young people with NT1 often experience widespread impairment in their psychological well-being, education, and social relationships [3,4,5]. The importance of a biopsychosocial approach in the treatment of children with NT1 is stressed [6,7,8]. Despite the urge for such a biopsychosocial approach, not much is known about the problems and possible interventions for children with NT1 at school.

Research that has been performed pointed out that youngsters with NT1 often face problems at school. Despite average IQ levels [9,10,11,12], young people with NT1 have more educational difficulties than youngsters with EDS alone or healthy controls [1,3,13]. Teachers have noticed several problems in children with NT1, including poor attention span, hyperactivity, distractibility, and cognitive underperformance. These school problems contribute significantly to a lower quality of life (QoL) in these individuals [3,14]. Adults with NT1 retrospectively evaluated their academic careers as more complex [13,15]. Moreover, prospective studies found that individuals with child- or adolescent-onset NT1 had lower educational levels, lower grading, and lower employment rate and income than healthy controls [16].

Treatment of NT1 in youngsters is focused on medication use [17] and behavioral approaches. Little research has been conducted on behavioral interventions for NT1. There is some evidence that daily napping is helpful in reducing EDS [18]. Another study that consisted of older adolescents and adults (please note that only 8.3% were younger than 17) reported some symptom improvement with sleep hygiene (taking regular naps and keeping a nocturnal schedule), environmental changes (such as temperature regulation) and physical exercise [19]. Keeping nocturnal schedules, sleep hygiene, balanced diet, and physical activities, are also mostly recommended in clinical-based guidelines [20]. To the best of our knowledge specific studies on interventions to decrease attention problems of children or adolescents with NT1 in classroom are lacking. It is unknown to which extent these behavioral interventions are implemented in the school-life of youngsters suffering from NT1. Specialized school workers in the Netherlands (‘the LWOE’, https://www.lwoe.nl/, accessed on 9 March 2023) help youngsters with epilepsy implementing interventions at school. Such a team of school workers has also start helping children with NT1. It is unknown whether such a team is beneficial for implementing behavioral interventions for NT1 at school.

This observational study aims to obtain more insight into specific problems encountered by children and adolescents with NT1 in school and into the implementation of potential interventions to increase school functioning. We also examined whether school support by a specialized team was associated with the higher implementation of these interventions within classrooms and better daytime functioning.

## 2. Methods

### 2.1. Participants

One hundred twelve children and adolescents diagnosed with NT1, aged from 4 to 19 years, were approached for participation in the study at three specialized sleep centers in the Netherlands with expertise in narcolepsy. The inclusion criteria of the study were being diagnosed with NT1 according to the ICSD3-criteria and attending primary or secondary school. Diagnoses were made according to ICSD-3 criteria [21]. Medical records were used to determine whether participants received school support from a specialized team of school workers (‘the LWOE’). Fifty-five individuals (49%) agreed to participate in the study. Young people were grouped according to school type; 18 participants were in primary school (‘children’), and 37 were in secondary school (‘adolescents’). Participants were recruited at different points after diagnosis. The median number of months since diagnosis was 33 for children and 29 for adolescents (Table 1). Most children and adolescents (>75%) received their diagnosis more than 1 year ago.

### 2.2. Procedure

Parents and adolescents gave informed consent after a full explanation of the study procedure. We reviewed medical charts to collect information regarding current medical treatment and comorbid conditions. Teachers were approached after parents and adolescents gave permission. They received paper-and-pencil questionnaires from the participating children and adolescents. They filled out the forms at home and returned them directly to the research institute. Adolescents, parents, and teachers completed questionnaires about problems experienced at school, school functioning, implementation of potential interventions at school, global functioning, and depressive symptom level. The study was conducted in accordance with the Declaration of Helsinki. The protocol was evaluated by the Medical Ethics Committee and judged as exempt from needing formal ethical approval due to its observational design.

### 2.3. Questionnaires

#### 2.3.1. Global Functioning in Children with Chronic Conditions (DISABKIDS)

The Quality of Life Questionnaire for Children with Chronic Conditions (DISABKIDS) is a questionnaire used to assess global functioning in children and adolescents with chronic diseases. A parent-version (proxy) was used for children in primary education. Adolescents in secondary schools completed a self-report version. The DISABKIDS (both proxy and self-report) contains 37 items, and answers are scored on a 5-point Likert scale. A high total score indicates better global functioning. The questionnaire has six sub-scales: Independence, Emotions, Social Inclusion, Social Exclusion, Physical Limitation, and Impact of Treatment. Independence assesses how much the disease impairs a child/adolescent and whether he/she can live an independent life. Emotion describes to what extent the condition causes worry or concern for the child/adolescent. Social inclusion measures the closeness and positivity of friends and family, whereas social exclusion measures the child’s/adolescent’s feelings of stigma. Physical Limitation assesses to what degree the disease limits the child/adolescent. The Impact of Treatment subscale measures the child’s/adolescent’s negative feelings about taking medication. Raw total and sub-scale scores are transformed into t-scores for further interpretation. Normative scores are available from children/adolescents with various chronic conditions, among which idiopathic epilepsy [22]. The Cronbach’s alpha for the different scales in our study varied. Some were poor [physical limitation (0.47)/social inclusion (0.59)]; others acceptable [independence (0.74)/social exclusion (0.75)/total scale (0.79)], and others good [emotion (0.84) treatment (0.85)].

#### 2.3.2. Depressive Symptoms

The Children’s Depression Inventory (CDI) is a self-reported questionnaire to measure depressive symptoms in children and adolescents. The questionnaire is validated for ages 7–17 [23,24]. Each item contains three possible answers and generates an item score ranging from 0 (=absence of symptom) to 2 (=symptom present). The sum of all twenty-seven item scores generates the raw total score. No age-specific reference values are available for the CDI.

#### 2.3.3. School Functioning

Children, adolescents, their parents, and teachers were asked to complete questionnaires to assess problems with NT1 encountered at school. A panel of professionals, consisting of neurologists, nurse practitioners, psychologists, and student counselors working with young people with NT1, developed the questionnaire.

Additionally, teachers answered open-ended questions about the three most critical problems they had noticed in their students with NT1. Two researchers (KJ and DH) independently categorized these answers, e.g., the answers ‘tiredness,’ ‘feeling fatigued,’ ‘overtiredness,’ and ‘lack of energy’ were all assigned to the category “tiredness/lack of energy.” These categories were compared and discussed until a consensus was reached. Thus, an overview of the most common problems experienced in the classroom was generated.

#### 2.3.4. School Interventions

Adolescents (≥12 years), parents (of children/adolescents <12), and their teachers answered questions about whether arrangements were made between the young person and teacher about adjustments at school. Fourteen different adjustments were considered (Appendix A).

Adolescents, parents, or teachers could answer whether they made agreements about these interventions by choosing one of the following answers: ‘yes’/’no’/’unknown.’ All questions that were answered with ‘yes’ were cumulated to calculate the total amount of arranged interventions for each child. Agreement between parents and teachers differed between poor and excellent for the separate interventions (Appendix A). The correlations between the total number of arranged interventions were adequate (Spearman rho parent/teacher: 0.44; adolescent/teacher: 0.68).

#### 2.3.5. The LWOE

The LWOE consists of a team of school counselors specialized in supporting young people with epilepsy and NT1 at school. They have regular appointments with young people with NT1, their parents, and teachers. They discuss problems young people with NT1 face at school, potential interventions for these problems, and evaluate the outcomes of these interventions. They help them to achieve more understanding of their problems from their teachers and classmates. Further, they discuss whether they receive appropriate education or whether further assistance or a specialized school is necessary to reach the full potential of the youngster.

### 2.4. Analyses

Young people’s scores on the DISABKIDS were compared with normative scores of young people with epilepsy with one-sample t-tests. Data of the DISABKIDS in both samples were normally distributed. An overview of problems encountered by young people with NT1 reported by the teachers was calculated. Two researchers independently grouped the answers into different categories. Cases of disagreement were discussed. Further, an overview was generated of the number of interventions in the classroom that were discussed by students and teachers. For the primary school children, we used parent, and teacher questionnaires. For the secondary school children, self-reports and teacher questionnaires were used. We then compared the results between young people who received school support to those who did not receive school support on global functioning, depressive symptom level, and a number of arranged interventions, using independent sample t-tests. The difference between both groups in napping at school and during the weekends was calculated with Fisher’s exact tests. When a whole questionnaire was not completed by a child/parent or teacher, these participants were excluded from the analyses concerning this questionnaire. Single missing items were coded in a separate category, “unknown.” Percentages were calculated while considering this category.

All analyses were performed with SPSS, version 23. A *p*-value of <0.05 was considered statistically significant.

## 3. Results

In total 18 children and 37 adolescents with NT1 participated. Eight of 18 (44%) children and 18 of 37 (48%) adolescents received school support from specialized school workers. Characteristics of the groups are shown in Table 1.

Young people with NT1 scored lower than those with epilepsy on the emotion, social exclusion, physical limitation scale, and total scores (Table 2).

The mean depression score of children with NT1 was 11.0 (SD = 9.1); that of adolescents was 7.2 (SD = 4.7). The clinical cut-off of the CDI of 16 has been recommended as an indication of a depressive disorder in all age categories [23]. Four of the twelve children (33%) and one of 36 adolescents (3%) who completed this questionnaire scored above this cut-off.

According to the parents, 4 out of 17 (25%) children napped daily at school, whereas 12 out of 17 (71%) napped daily during the weekends. According to parents, 23 of 36 (64%) of the adolescents napped daily during school, and 29 of 36 (81%) napped daily during the weekends. One parent did not answer these questions. According to the adolescents themselves, 22 out of 37 (59%) napped daily during school, and 27 of 37 (73%) napped daily during the weekends.

School reports were obtained for 16 children (89%) and 24 adolescents (65%). The most frequent problems reported by the teachers of the 16 children with NT1 were tiredness/lack of energy, concentration problems, problems with social interaction, and living in their own world/autistic-like behavior (Table 3). The most frequent problems reported by the teachers of adolescents with NT1 were tiredness/lack of energy, concentration problems, and feeling anxious/insecure or auditory information processing problems.

According to the teachers, half the children at primary school arranged to take naps during class, and about one-third had a place to sleep outside the classroom (Table 4). The most common was to discuss school excursions, and a minority arranged to obtain extra time for exams. Teachers reported a higher level of arrangement than parents did.

Three-quarters of the adolescents at secondary school had a place to nap outside the classroom, and most arranged to take naps during class (Table 4). Extra time for taking a test, discussing school excursions, and adaptations during sports classes were common interventions for adolescents.

### School Support

Children and adolescents who received school support from a specialized team reported significantly more interventions during class than children who did not receive school support (see Table 5). Children who received school support napped more frequently during school, i.e., 4 out of 7 who received school support napped, whereas 0 out of 9 who did not receive school report napped at school (Fisher’s exact test: *p* = 0.02). The same holds for adolescents, i.e., 14 out of 17 who received school support napped, whereas 7 out of 17 who did not receive school reports did (Fisher’s exact test: *p* = 0.03). Children who received school support did not differ in whether they napped on a regular basis at home during the weekends; 5 out of 8 who received school reports napped at home during the weekends, and 6 out of 9 who did not receive school support did (Fisher’s exact test: *p* = 0.63). The same was true for adolescents, 15 out of 17 who received school reports napped at home during the weekends, and 12 out of 17 who did not receive school support did (Fisher’s exact test: *p* = 0.39). Receiving school support was not associated with better global functioning or fewer depressive symptoms (Table 5).

## 4. Discussion

This observational study suggests that children and adolescents with NT1 experience a diversity of issues at school. The most common problems at school, according to teachers, were concentration problems and feeling fatigued. Some interventions were common, whereas most were only used by a minority of the children with NT1. Receiving school support from a specialized team was associated with more interventions being implemented within the classroom and with napping on a regular basis at school.

According to the teachers, the most commonly reported problems at school were concentration problems and feeling fatigued at school. These symptoms have also been found to be the most burdensome symptoms, according to adults with NT1 [19]. Adaptations within the classroom might be helpful to diminish these problems. Some school interventions were common, such as arranging school excursions and taking a nap at school. Other interventions, such as those aimed at increasing attention (e.g., drawing, walking and eating during instructions, and the use of audiobooks), were only used by a minority of the young people. The low rate of interventions to increase attention leaves room for improvement. Children with NT1 might, for example, benefit from organizational skills interventions that have been developed and found to be beneficial in children with attention-deficit/hyperactivity disorder [25].

Secondary school teachers reported more implemented interventions than primary school teachers did. For example, only 31% of the primary school teachers reported that they made arrangements about a place to nap inside the school, whereas 75% of the secondary school teachers did. It might be that some interventions have been less applicable to primary school children since school days for primary school children are shorter, they have less homework, and schools are often located closer to children’s homes. The difference between the number of children that were daily napping at school and daily napping at home during the weekends was also large. Twenty-five percent of children with NT1 napped during school days, whereas 71% napped during the weekends. The reasons for this difference are unclear. Future research is needed to examine why behavioral interventions are not always implemented at school. For example, are facilities not available, is there a lack of understanding from classmates or teachers to implement these interventions, or do adolescents feel ashamed to make use of these interventions? We might learn from adherence research in adolescents with diabetes type 1 since these adolescents also have to actively manage their disease in the classroom [26,27].

The current study shows that young people who received school support from specialized teams of school workers made more arrangements for interventions than children who did not receive school support. Interestingly, young people who received school support napped more frequently at school but did not nap more at home during the weekends. In this observational study, however, receiving school support was not associated with better school functioning, better global functioning, or lower depressive symptom level. This might be related to the cross-sectional observational nature of this study. Whether or not the children received school support was not randomly determined but was determined by the clinician in accordance with the child and the parent. Children who experienced more school problems or problems with global functioning might therefore have been more prone to receive school support. This might have weakened potential associations between school interventions and global functioning. Moreover, since school support only focuses on school problems, a broader intervention program for children with NT1 might be necessary to have an effect on the global functioning level. Strong points of this study include the multi-informant design. Information was obtained from young people, their parents, and their teachers. We were, therefore, able to examine school functioning and interventions from different perspectives. Although perspectives were different, given the sometimes low correlations, the percentages of implemented interventions were largely comparable in different groups, which enlarges the robustness of our findings. Finally, we used extensive questionnaires and have a broad overview of youngsters functioning in and outside school.

Some limitations of this study should also be mentioned. The number of participants was low—but NT1 is a relatively rare disease. The participation rate of almost 50% was somewhat low but in line with dropping participation rates in general [28]. We used paper-and-pencil questionnaires. This might have reduced the number of received responses and diminished the generalizability of our findings. Additionally, participants were not prompted to choose an answer, which resulted in open fields or questions indicated as “unknown.” Future qualitative studies in which participants, parents, and teachers are interviewed about school functioning might help in getting additional information about the specific needs of children and adolescents with NT1 and ways to ease the implementation of interventions. Another limitation is the cross-sectional and observational study design. We were not able to study the effect of school support on functioning in young people with NT1. Longitudinal studies with assessment of school functioning before and after school support are necessary to obtain more insight into school support’s effects on young people with NT1.

This study shows that there is room for improvement in the implementation of interventions for children with NT1 in the classroom, such as taking naps during school time. Although we should be careful because of the observational character of our study, our study suggests that a team of specialized school workers might be helpful in improving this implementation. We further believe that awareness of teachers for problems children and adolescents with NT1 experience at school, especially overtiredness and concentration problems are important. The teacher might think along with students and stimulate students to use these interventions during school time. Further research is necessary to develop a specific behavioral intervention to increase attention in youngsters with NT1. For example, it might be good to examine whether CBT-based interventions developed to decrease attention problems in children with attention deficit/hyperactivity disorder [29,30] are also helpful for youngsters with NT1.

## 5. Conclusions

Children and adolescents with NT1 are at risk of experiencing problems inside the school. Feeling fatigued and experiencing concentration problems are common at school. Potential interventions were only partially implemented, especially since the number of primary school children that napped during school time (25%) was low. School support by specialized teams in this observational study was associated with better implementation of these interventions and specifically more napping at school, but not with better school functioning or better quality of life. Longitudinal interventional research is necessary to get more insight into the benefit of behavioral interventions for narcolepsy at school and into ways to implement them.

## Figures and Tables

**Table 1 ijerph-20-05175-t001:** Sample characteristics.

	Children (n = 18)	Adolescents (n = 37)
Age (years)	mean 10.7 (SD 1.4) range: 7–12	mean 15.4 (SD 1.8) range: 13–19
Male	55.6% (10/18)	48.6% (18/37)
Co-morbid medical or psychiatric conditions *	47% (8/17)	32% (11/34)
Medication use *	82% (14/17)	94% (32/34)
Type of medication	Methylphenidate 8/17 Dexamphetamine 3/17 Modafinil 4/17 Clomipramine 1/17 Xyrem 2/17 Concerta 1/17 Pitalosant 0/17	Methylphenidate 14/30 Dexamphetamine 5/30 Modafinil 3/30 Clomipramine 9/30 Xyrem 14/30 Concerta 2/30 Pitalosant 1/30
Time since diagnosis (months) *	M 33 (IQR: 11–43)	M 29 (IQR: 12–44)
School support	44.4% (8/18)	51% (19/37)
Ethnicity	16 Caucasian 1 Asian	2 mixed-African 1 Mixed South-American 27 Caucasian

* Medical information was not available for all participants; SD = standard deviation; M = median, IQR = interquartile range.

**Table 2 ijerph-20-05175-t002:** Comparison of global functioning to norm scores of young people with epilepsy.

	NT1 Primary School (Proxy-Report)	NT1 Secondary School (Self-Report)
DISABKIDS independence	64.6 (SD 18.6)	75.5 (SD 12.1)
DISABKIDS Emotion	51.2 (SD 15.8) *	66.1 (SD 18.2) *
DISABKIDS Social exclusion	62.9 (SD 18.1) *	74.7 (SD 16.7) *
DISABKIDS Social inclusion	59.7 (SD 14.9)	72.5 (SD 12.9) *
DISABKIDS physical limitation	46.1 (SD 17.2) *	55.0 (SD 16.8) *
DISABKIDS treatment	71.4 (SD 23.9)	78.0 (SD 20.6)
DISABKIDS total	57.5 (SD 14.9) *	70.0 (SD 11.9) *

* significantly different from norm scores of children with epilepsy [22] in a one sample t-test; data in both cohorts were about normally distributed; higher scores indicate higher global functioning; numbers are presented as mean (SD).

**Table 3 ijerph-20-05175-t003:** The three most frequently reported problems by teachers in patients with NT1.

	Children (n = 16)	Adolescents (n = 24)
Concentration problems	10 (63%)	15 (63%)
Tiredness/lack of energy	9 (56%)	15 (63%)
Problems with peers	4 (25%)	2 (8%)
Autistic-like behavior/living in own world	4 (25%)	1 (4%)
Seems absent/auditory information processing problems	3 (19%)	4 (17%)
Hyperactive/impulsive behavior	3 (19%)	1 (4%)
Sleepiness	2 (13%)	2 (8%)
Anxiety or mood problems	2 (13%)	4 (17%)
Problems with teachers	1 (6%)	1 (4%)
Slow work pace	1 (6%)	1 (4%)
Planning and organizing skills	1 (6%)	1 (4%)
School absenteeism	1 (6%)	1 (4%)
Memory problems	0 (0%)	1 (4%)

Please note that the answer categories were grouped by two researchers independently from the open-ended answers of the teachers.

**Table 4 ijerph-20-05175-t004:** Arrangements about adjustments made at primary and secondary school according to teachers.

	Primary School (n = 16)			Secondary School (n = 24)		
	Yes	No	N.a./Unknown	Yes	No	N.a./Unknown
Discussions about school excursions	11 (69%)	3 (19%)	2 (13%)	17 (71%)	2 (8%)	5 (21%)
Naps during school	8 (50%)	5 (31%)	3 (19%)	14 (58%)	6 (25%)	4 (17%)
Extra time for tests	7 (44%)	5 (31%)	4 (25%)	12 (50%)	5 (21%)	7 (29%)
Availability of a place to nap outside the classroom	5 (31%)	6 (38%)	5 (31%)	18 (75%)	2 (8%)	4 (17%)
Taking a nap before or after a test	4 (25%)	8 (50%)	4 (25%)	8 (33%)	5 (21%)	9 (46%)
Adaptations during sport class	5 (31%)	9 (56%)	2 (13%)	10 (42%)	4 (17%)	10 (42%)
Naps during breaks	4 (25%)	7 (44%)	5 (31%)	5 (21%)	10 (42%)	9 (38%)
Extra guidance when information missed due to naps	2 (13)	8 (50%)	6 (38%)	3 (13%)	8 (33%)	13 (54%)
Day planning about timings for test and naps	2 (13%)	6 (38%)	8 (50%)	7 (29%)	4 (17%)	13 (54%)
Allowed to eat when receiving instructions	2 (13%)	9 (56%)	5 (31%)	10 (42%)	5 (21%)	9 (38%)
Allowed to draw when receiving instructions	1 (6%)	12 (75%)	3 (19%)	5 (21%)	1 (4%)	18 (75%)
Allowed to walk when receiving instructions	1 (6%)	9 (56%)	6 (38%)	5 (21%)	5 (21%)	14 (58%)
Adjustments of the amount of homework or tests	1 (6%)	8 (50%)	7 (44%)	5 (21%)	7 (29%)	12 (50%)
Use of audio books	0	11 (69%)	5 (31%)	1 (4%)	13 (54%)	10 (42%)

**Table 5 ijerph-20-05175-t005:** The association between school support and school functioning, general functioning and the number of interventions implemented.

	No School Support		School Support by Specialized Teams	
	Children (Parent Report, n = 10)	Adolescents (Self-Report, n = 19)	Children (Parent Report, n = 8)	Adolescents (Self-Report, n = 18)
Age (years)	10.7 (SD 1.4)	15.3 (SD 1.7)	10.8 (SD 1.4)	15.4 (SD 2.0)
Male	4/10	10/19	6/8	8/18
Depressive symptoms (CDI)	11.3 (SD 10.6)	6.9 (SD 4.5)	10.6 (SD 7.5)	7.4 (SD 5.2)
DISABKIDS-total	57.0 (SD 17.6)	71.1 (SD 10.4)	57.9 (SD 13.2)	69.2 (SD 13.3)
DISABKIDS-Emotion	47.9 (SD 16.9)	69.4 (SD 19.1)	55.4 (SD 14.2)	63.5 (SD 17.5)
DISABKIDS-Social inclusion	64.2 (SD 15.2)	70.5 (SD 9.1)	54.2 (SD 13.4)	76.8 (SD 11.3)
DISABKIDS-Social exclusion	65.4 (SD 18.7)	78.3 (SD 13.8)	59.9 (SD 18.1)	71.9 (SD 18.6)
DISABKIDS-Physical limitation	42.9 (SD 19.4)	58.3 (SD 21.2)	50.0 (SD 14.3)	51.9 (SD 11.1)
DISABKIDS-Treatment	67.3 (SD 23.3)	80.8 (SD 21.6)	76 (SD 25.6)	76.3 (SD 20.4)
DISABKIDS-Independence	63.8 (SD 19.5)	77.1 (SD 10.8)	74.3 (SD 13.1)	74.3 (SD 13.1)
Number of interventions	1.0 (SD 1.0) *	3.2 (SD 2.3) *	4.1 (SD 1.9) *	5.2 (SD 2.7) *
naps during weekdays	0/9 **	7/17 **	4/7 **	14/17 **
naps during the weekends	6/9	12/17	5/8	15/17

* The difference between the group that received and did not receive school support by a specialized team was statistically significant according to an independent sample t-test, *p* < 0.05; ** The difference between the group that received and did not receive school support by a specialized team was statistically significant according to a Fisher exact test, *p* < 0.05; all 18 children completed the CDI, DISABKIDS; the CDI was completed by 36 adolescents, the DISABKIDS by 34; information about naps was available for 9 children and 17 adolescents who did not receive school support, and from 7 (weekdays) to 8 (weekends) children and 17 adolescents who received school support.

## Data Availability

The data presented in this study are available on request from the corresponding author. The data are not publicly available due to privacy reasons.

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
