# Peer review of "School Problems and School Support for Children with Narcolepsy: Parent, Teacher, and Child Reports"

_ijerph, 2023, doi:10.3390/ijerph20065175_

Round 1

Reviewer 1 Report

Dear authors, thank you for giving me the opportunity to review your manuscript: “School problems and school support for children with narcolepsy: parent, teacher and child reports”. 

I send below some comments to improve the manuscript.

The manuscript does not present the references according to the latest APA norms (7th edition).

Introduction:

  • From lines 65-70: please provide references.
  • The introduction needs further development on intervention in children with narcolepsy.

Methods

-  Please provide the inclusion criterium to participate in the study.

Discussion:

  • Lines 283-290 should be discussed with bibliographic support.
  • The authors should develop the discussion further.
  • The authors should add implications for practice.

Author Response

Response to comments of reviewer 1

Dear authors, thank you for giving me the opportunity to review your manuscript: “School

problems and school support for children with narcolepsy: parent, teacher and child reports”.

I send below some comments to improve the manuscript.

We would like to thank the reviewer for the useful comments and will respond to them below.

The manuscript does not present the references according to the latest APA norms (7th edition).

We have now adjusted the references to the APA 7 norms.

Introduction:

From lines 65-70: please provide references.

Unfortunately, there are no previous publications available about the LWOE. We have now added a link to their website, although we realize that it is in Dutch and focusses more on epilepsy than on narcolepsy.

The introduction needs further development on intervention in children with narcolepsy.

We have added more information on the intervention for children with NT1. We changed:

“There is some evidence that daily napping is helpful in reducing EDS (Mullington & Broughton, 1993). Other behavioral interventions for NT1, such as keeping nocturnal schedules, sleep hygiene, balanced diet, and physical activities, are based on clinical-based guidelines(Blackwell et al., 2022).”

Into:

“There is some evidence that daily napping is helpful in reducing EDS (Mullington & Broughton, 1993).  Another study that consisted of older adolescents and adults (please note that only 8.3% was younger than 17) reported some symptom improvement with sleep hygiene (taking regular naps and keeping a nocturnal schedule), environmental changes (such as temperature regulation) and physical exercise (Maski et al., 2017). Keeping nocturnal schedules, sleep hygiene, balanced diet, and physical activities, are also mostly recommended in clinical-based guidelines (Blackwell et al., 2022). To the best of our knowledge specific studies on interventions to decrease attention problems of children or adolescents with NT1 in classroom are lacking.”

Methods

Please provide the inclusion criterium to participate in the study.

We have added the inclusion criteria to the method section:

“The inclusion criteria of the study were being diagnosed with NT1 according to the ICSD3 criteria and attending primary or secondary school”

Discussion:

Lines 283-290 should be discussed with bibliographic support.

These lines represent the results of the current study. We now make this more clear and changed:

“Young people who received school support by specialized teams of school workers made more arrangements about interventions than children who did not receive school support.”

Into:

“The current study shows that young people who received school support by specialized teams of school workers made more arrangements about interventions than children who did not receive school support.”

The authors should develop the discussion further.

We have extended the discussion with more information on the result that school support was not associated with better quality of life and added implication for clinical practice, see also the next point.

“In this observational study, however, receiving school support was not associated with better school functioning, better global functioning or lower depressive symptom level. This might be related to the cross-sectional observational nature of this study. Whether or not the children received school support was not randomly determined, but was determined by the clinician, in accordance with the child and the parent. Children who experienced more school problems or problems with global functioning might therefore have been more prone to receive school support. This might have weakened potential associations between the school interventions and global functioning. Moreover, since school support only focuses on school problems, a broader intervention program for children with NT1 might be necessary to have effect on global functioning level”

The authors should add implications for practice.

Although we think we have to be careful with drawing too strong conclusions upon this observational study. We have now added some implication for clinical practice:

“This study shows that there is room for improvement for the implementation of interventions for children with NT1 in classroom, such as taking naps during school time. Although we should be careful because of the observational character of our study, our study suggests that a team of specialized school workers might be helpful to improve this implementation. We further believe that awareness of teachers for problems children and adolescents with NT1 experience at school, and especially overtiredness and concentration problems is important. Teacher might think along with students and stimulate students to use these interventions during school time. Further research is necessary to develop specific behavioral intervention to increase attention in youngsters with NT1. For example, it might be good to examine whether CBT-based interventions developed to decrease attention problems in children with ADHD (Anastopoulos et al., 2021; Solanto & Scheres, 2021) are also helpful for youngsters with NT1.”

Reviewer 2 Report

The study examines problems faced by children with type 1 narcolepsy (NT1) at school and investigates potential interventions for these problems. Children and adolescents with NT1 from three Dutch sleep-wake centers were recruited. The findings include identifying problems that children with NT1 face at school as well as a lack of interventions to assist these children with NT1. It was generally well written, with adequate reasoning to support the hypotheses. The methods were conducted well, using appropriate data, and the results were reported clearly and accurately. There are, however, some issues that could be addressed to strengthen the contributions of this paper.

I like the introduction, as it provided a good overview of the context and relevant literature on the topic.

It is not clear how you define the groups in Table 2.  The author made a comparison in the paragraph between children and adolescents who received school support and those who did not. The column header indicates a comparison between NT1 vs Norm scores for epilepsy, which means the comparison is not between the intervention  group vs the control group? Also, it seems  the values of n are missing.

A question related to my previous point is how the demographic characteristics of the groups presented in Table 2 differ. Because the sample size is small, the author may consider a rank sum test due to the possibility of your data having a non-normal distribution.

In table 5, it seems the authors did not include a comparison of naps as they described in the passages.

The insignificant associations between interventions and global functioning should be discussed in depth. Why this program does not impact these aspects compared to previous findings and how to improve this intervention can be discussed.

Author Response

Response to comments of reviewer 2

The study examines problems faced by children with type 1 narcolepsy (NT1) at school and investigates potential interventions for these problems. Children and adolescents with NT1 from three Dutch sleep-wake centers were recruited. The findings include identifying problems that children with NT1 face at school as well as a lack of interventions to assist these children with NT1. It was generally well written, with adequate reasoning to support the hypotheses. The methods were conducted well, using appropriate data, and the results were reported clearly and accurately. There are, however, some issues that could be addressed to strengthen the contributions of this paper.

I like the introduction, as it provided a good overview of the context and relevant literature on the topic.

We would like to thank the reviewer for these nice words and the useful feedback. We believe it helped to improve the manuscript.

 It is not clear how you define the groups in Table 2.  The author made a comparison in the paragraph between children and adolescents who received school support and those who did not. The column header indicates a comparison between NT1 vs Norm scores for epilepsy, which means the comparison is not between the intervention  group vs the control group? Also, it seems  the values of n are missing.

It is true that we compared the children of our study to the norm scores of epilepsy. We understand that it is perhaps misleading to put these numbers in the Table, since they have not been produced by our study. Therefore, we now only show the numbers on Quality of Life that we found in our own study and only indicate whether these scores significantly differed from the available norm scores of children with epilepsy.

 A question related to my previous point is how the demographic characteristics of the groups presented in Table 2 differ. Because the sample size is small, the author may consider a rank sum test due to the possibility of your data having a non-normal distribution.

The demographic characteristics of the reference group differ, since the reference data are collected in seven countries across Europe. Moreover, they consist of youngsters suffering from epilepsy. However, to the best of our knowledge these are the only available data to compare our data to. Since we have now raw data of the norm data, we are not able to do a rank sum test. However, since our data and the data in the reference study were about normally distributed, we believe a one sample t-test is justified. 

In table 5, it seems the authors did not include a comparison of naps as they described in the passages.

It is true that we only presented these numbers in the text to prevent overlap between the numbers in text and tables. We have now added these numbers to Table 5.

 The insignificant associations between interventions and global functioning should be discussed in depth. Why this program does not impact these aspects compared to previous findings and how to improve this intervention can be discussed.

We agree and changed:

“In this observational study, however, receiving school support was not associated with better school functioning, better global functioning or lower depressive symptom level. This might be related to the cross-sectional observational nature of this study, as children with more problems may have received more school support.”

Into

“In this observational study, however, receiving school support was not associated with better school functioning, better global functioning or lower depressive symptom level. This might be related to the cross-sectional observational nature of this study. Whether or not the children received school support was not randomly determined, but was determined by the clinician, in accordance with the child and the parent. Children who experienced more school problems or problems with global functioning might therefore have been more prone to receive school support. This might have weakened potential associations between the school interventions and global functioning. Moreover, since school support only focuses on school problems, a broader intervention program for children with NT1 might be necessary to sort effects on global functioning level”

Further, we have extended the article with the clinical implications and suggestion for improvement of the interventions:

“This study shows that there is room for improvement for the implementation of interventions for children with NT1 in classroom, such as taking naps during school time. Although we should be careful because of the observational character of our study, our study suggests that a team of specialized school workers might be helpful to improve this implementation. We further believe that awareness of teachers for problems children and adolescents with NT1 experience at school, and especially overtiredness and concentration problems is important. Teacher might think along with students and stimulate students to use these interventions during school time. Further research is necessary to develop specific behavioral intervention to increase attention in youngsters with NT1. For example, it might be good to examine whether CBT-based interventions developed to decrease attention problems in children with ADHD (Anastopoulos et al., 2021; Solanto & Scheres, 2021) are also helpful for youngsters with NT1.”

Anastopoulos, A. D., Langberg, J. M., Eddy, L. D., Silvia, P. J., & Labban, J. D. (2021). A randomized controlled trial examining CBT for college students with ADHD. J Consult Clin Psychol, 89(1), 21-33. https://doi.org/10.1037/ccp0000553

Blackwell, J. E., Kingshott, R. N., Weighall, A., Elphick, H. E., & Nash, H. (2022). Paediatric narcolepsy: a review of diagnosis and management. Arch Dis Child, 107(1), 7-11. https://doi.org/10.1136/archdischild-2020-320671

Maski, K., Steinhart, E., Williams, D., Scammell, T., Flygare, J., McCleary, K., & Gow, M. (2017). Listening to the Patient Voice in Narcolepsy: Diagnostic Delay, Disease Burden, and Treatment Efficacy. J Clin Sleep Med, 13(3), 419-425. https://doi.org/10.5664/jcsm.6494

Mullington, J., & Broughton, R. (1993). Scheduled naps in the management of daytime sleepiness in narcolepsy-cataplexy. Sleep, 16(5), 444-456. https://doi.org/10.1093/sleep/16.5.444

Solanto, M. V., & Scheres, A. (2021). Feasibility, Acceptability, and Effectiveness of a New Cognitive-Behavioral Intervention for College Students with ADHD. J Atten Disord, 25(14), 2068-2082. https://doi.org/10.1177/1087054720951865
